# Associations between Daily Movement Distribution, Bone Structure, Falls, and Fractures in Older Adults: A Compositional Data Analysis Study

**DOI:** 10.3390/ijerph18073757

**Published:** 2021-04-03

**Authors:** Ana Moradell, Irene Rodríguez-Gómez, Ángel Iván Fernández-García, David Navarrete-Villanueva, Jorge Marín-Puyalto, Jorge Pérez-Gómez, José Gerardo Villa-Vicente, Marcela González-Gross, Ignacio Ara, José Antonio Casajús, Alba Gómez-Cabello, Germán Vicente-Rodríguez

**Affiliations:** 1GENUD (Growth, Exercise, NUtrition and Development) Research Group, Universidad de Zaragoza, 50009 Zaragoza, Spain; amoradell@unizar.es (A.M.); angelivanfg@unizar.es (Á.I.F.-G.); dnavarrete@unizar.es (D.N.-V.); jmarinp@unizar.es (J.M.-P.); joseant@unizar.es (J.A.C.); agomez@unizar.es (A.G.-C.); 2Agrifood Research and Technology Centre of Aragón, IA2, CITA—Universidad de Zaragoza, 50009 Zaragoza, Spain; 3Exercise and Health in Special Population Spanish Research Net (EXERNET), 50009 Zaragoza, Spain; marcela.gonzalez.gross@upm.es; 4Faculty of Health and Sport Science (FCSD), Department of Physiatry and Nursing, University of Zaragoza, Ronda Misericordia 5, 22001 Huesca, Spain; 5GENUD Toledo Research Group, University of Castilla-La Mancha, 45071 Toledo, Spain; irene.rodriguez@uclm.es (I.R.-G.); ignacio.ara@uclm.es (I.A.); 6Biomedical Research Networking Center on Frailty and Healthy Aging (CIBERFES), 28029 Madrid, Spain; 7Department of Physiatry and Nursing, Faculty of Health, University of Zaragoza, 50009 Zaragoza, Spain; 8HEME (Health, Economy, Motricity and Education) Research Group, Faculty of Sport Science, University of Extremadura, 10003 Cáceres, Spain; jorgepg100@unex.es; 9VALFIS Research Group, Department of Physical Education and Sport, Institute of Biomedicine (IBIOMED), University of León, 24007 León, Spain; jg.villa@unileon.es; 10ImFINE Research Group, Department of Health and Human Performance, Faculty of Physical Activity and Sport Sciences-INEF, Polytechnic University of Madrid, 28040 Madrid, Spain; 11Centro de Investigación Biomédica en Red de Fisiopatología de la Obesidad y Nutrición (CIBERObn), 28029 Madrid, Spain; 12Centro Universitario de la Defensa, 50090 Zaragoza, Spain

**Keywords:** bone mineral density, elderly, moderate-to-vigorous physical activity, sedentary time

## Abstract

With aging, bone density is reduced, increasing the risk of suffering osteoporosis and fractures. Increasing physical activity (PA) may have preventive effects. However, until now, no studies have considered movement behaviors with compositional data or its association to bone mass and structure measured by peripheral computed tomography (pQCT). Thus, the aim of our study was to investigate these associations and to describe movement behavior distribution in older adults with previous falls and fractures and other related risk parameters, taking into account many nutritional and metabolic confounders. In the current study, 70 participants above 65 years old (51 females) from the city of Zaragoza were evaluated for the EXERNET-Elder 3.0 project. Bone mass and structure were assessed with pQCT, and PA patterns were objectively measured by accelerometry. Prevalence of fear of falling, risk of falling, and history of falls and fractures were asked through the questionnaire. Analyses were performed using a compositional data approach. Whole-movement distribution patterns were associated with cortical thickness. In regard to other movement behaviors, moderate-to-vigorous PA (MVPA) showed positive association with cortical thickness and total true bone mineral density (BMD) at 38% (all *p* < 0.05). In addition, less light PA (LPA) and MVPA were observed in those participants with previous fractures and fear of falling, whereas those at risk of falling and those with previous falls showed higher levels of PA. Our results showed positive associations between higher levels of MVPA and volumetric bone. The different movement patterns observed in the groups with a history of having suffered falls or fractures and other risk outcomes suggest that different exercise interventions should be designed in these populations in order to improve bone and prevent the risk of osteoporosis and subsequent fractures.

## 1. Introduction

The beneficials effects of physical activity (PA) are well known across the human lifespan [1]. PA prevents multiple noncommunicable diseases, and it is positively associated with mental health, quality of life, and well-being during elderhood [2]. Moreover, PA also improves body composition as it maximizes bone peak mass during the first three decades of life and minimizes the age-related loss of bone mass and density, reducing the risk of osteoporosis and its subsequent fractures, which have commonly led to dependence and an increased risk of frailty [3,4]. PA may also increase muscle mass, as well as physical function, leading to a reduction on the risk of falls [5], which is strongly related with the abovementioned bone fractures.

In contrast, having a sedentary lifestyle with an excess of sitting time has negative consequences for health, reversing even those PA benefits mentioned above [6,7]. Although some authors have reported associations between poor bone health and sedentarism in early stages of life [8,9], literature about what happens in older adults is still scarce and, therefore, more evidence is needed. Strategies focused on reducing sedentary behaviors (SB) and increasing PA have been common in order to reverse the fear of falling, decrease fall rates, and reduce the number of bone fractures [10,11,12,13]. However, movement behavior patterns should be studied as a whole in order to understand differences between older adults who have a fear of falling and those who do not, fallers and non-fallers, and those who are at risk of falling or not, which would make it possible to design specific interventions in order to prevent future fractures. In addition to muscle mass, which may be associated with healthier bone properties [14,15], other behavioral or biological variables such as nutritional ones (calcium or alcohol intake) and serum vitamin D levels have been associated with bone improvement alone or combined with PA or exercise [16].

A 24-hour period is finite, and more time spent in one behavior necessarily decreases the time spent in another behavior. New tendencies about the study of movement behaviors consider them as a whole [17]. Compositional data analyses (CoDA) deals with the fundamental nature of movement behavior data, which are intrinsically compositional [18,19], making it possible to eliminate collinearity problems and deal with the codependence between time spent in different movement behaviors [18,19].

In this context, Rodriguez-Gómez et al. were the first authors to establish the relationships of the distributed PA behaviors with bone variables measured by dual-energy X-ray absorptiometry (DXA) in older individuals in several studies [20,21,22,23]. However, the use of other devices assessing volumetric true bone mineral density (BMD) or bone structural parameters such as peripheral Quantitative Computed Tomography (pQCT) has not been applied. Although DXA has lower dose of radiation, it estimates areal BMD, and these bidimensional measurements could overestimate BMD in larger bone. Thus, a volumetric evaluation, as well as the assessment of other variables related to bone structure, architecture, and strength provided by pQCT may contribute to a deeper knowledge of the relationship between distribution of movement behaviors and bone health. pQCT allows for the analysis of peripheral body segments, namely the tibia and radius. However, in older and frail people, the measurement of the tibia is less difficult and the quality of the measurements is superior. Furthermore, although the tibial bone may not be as related to osteoporosis as other parts of the skeleton, it may be more influenced by PA-induced muscle forces.

To the best of our knowledge, no other studies have investigated the relationship between movement behavior distribution and bone mass and structure evaluated by pQCT in older individuals. The main aims of this study were: (1) To identify how the movement behaviors profile is associated with bone mass and structure in older adults, and (2) to investigate the relationship between movement behaviors and previous falls and fractures, the fear of falling, and the risk of suffering falls.

## 2. Materials and Methods

### 2.1. Sample and Study Design

This was a cross-sectional study carried out with data from the initial sample of the EXERNET-Elder 3.0 project (2018). Briefly, this study aimed to improve physical function of frail and pre-frail older adults by a multicomponent exercise program. Initially, 123 participants were recruited from 4 health care centers and 3 nursing homes from the city of Zaragoza (Spain). Health care professionals from each center selected potential participants and derived them to the research team. Once the participants arrived, researchers confirmed that they met the inclusion criteria: Above 65 y, no diagnosis of cancer or dementia, and a score below 4 in the Short Physical Performance Battery [24]. For this report, only those who completed pQCT assessments and accelerometry records were included in the analyses.

A structured questionnaire was used to collect personal information and other health outcomes from individuals. Afterward, researchers performed body composition analysis (including pQCT) and other measurements to collect bone possible confounders, such as physical fitness measurements and a fasting blood sample test. The whole methodology has been previously described in detail elsewhere [25].

### 2.2. Ethics Statement

Oral and written information about the aims, possible benefits, and risks derived from participation in this study were given to participant. Participants needed to complete the written informed consent to be included in the study.

Study protocol was approved by the Hospital Universitario Fundación de Alcorcón (16/50) and registered inclinicaltrials.gov (reference number: NCT03831841).

All the study was according to the Helsinki Declaration of 1961 revised in Fortaleza (2013) [26] and the current legislation of human clinical research of Spain (Law 14/2007).

### 2.3. Peripheral Quantitative Computed Tomography (pQCT)

pQCT measurements were performed using a Stratec XCT-2000 L pQCT scanner (Stratec Medizintechnik, Pforzheim, Germany). A quality control was undertaken to ensure the measurements were correct.

The nondominant tibia was selected for the measurements. The reference line on the distal end of the tibia was chosen manually by trained researchers. The measurement sites were located by a distance corresponding to 4% (distal tibia), 38% (diaphyseal tibia), and 66% (largest calf perimeter) of the tibia length as previously described [27]. Bone parameters considered were total bone mineral content (Tt.BMC), total bone area (Tt.Ar), total true BMD (Tt.BMD) (all of them at 4% and 38% of the tibia), trabecular BMD at 4% of the tibia (Tb.BMD), cortical BMD (Ct.BMD), cortical bone thickness (Ct.Th) at 38% of the tibia length, and muscle area at 66%. Bone strength was established with respect to resistance to torsion (polar stress strain index in mm^3^ (SSIp)), bending (measured at 38%), and fracture load X (N) (measured at 66%), with respect to the X-axis, as described elsewhere [28].

A quality control of the pQCT measurements was performed by the same trained researcher each time (A.M.). The images obtained were evaluated, classifying them in a gradual scale from 1 to 5 points: 1 for those “perfect” without any movement, and 5 for those “impossible to use” with too much movement during the scan test, following the indications of Blew et al. [29]. Images scored from 4 to 5 were excluded as they were not considered valid to use [29].

### 2.4. Fear and Risk of Falling, Falls, and Fractures

Questions about fear of falling, falls suffered during last year, and their consequences were designed ad-hoc and added to the general questionnaire. Number of fractures during the last 10 years was also asked. The risk of falls was assessed through a scale from The Vivifrail instrument [30], which included time-up and go test [31], the walking speed test in 6 m [32], and the following two questions: (1) Have you been medically diagnosed of cognitive decline? and (2) have you had 2 or more falls during the last year? (or 1 with medical assistance). Demonstrating poor physical performance or answering one of those questions with “yes” answer led the participant to be in the “at-risk group.”

### 2.5. Physical Activity and Sedentary Behaviors

PA was monitored with wrist-worn triaxial accelerometers (GENEActiv, Activinsights Ltd., Cambridge, UK) that collected data for 1 week at a frequency of 10 Hz, which is sufficient to classify daily activities, according to Zhang et al. [33]. Elders wore the device on the nondominant wrist for 7 consecutive days, including 2 weekend days. The accelerometer is waterproof, so participants did not have to remove it for anything. Only participants with at least 4 valid days including at least 480 min (8 h/day) of wearing could be included. Non-wear time detection was evaluated in blocks of 30 consecutive minutes following the methods described by van Hees et al. [34]. Triaxial data was condensed in 1 vector, calculating the Euclidean norm minus 1 to isolate human movement from gravitational acceleration [34], and aggregated into 60 s epochs. Each epoch was classified as either SB, LPA, or MVPA time, according to previously defined cut-off points [35] which are specific for this population and accelerometer location and that have been designed to optimize both sensitivity and specificity of classification [36]. In order to differentiate sleep time (ST) from inactivite periods during the day, a sleep period time window was detected [37].

### 2.6. Body Composition Measurements and Anthropometrics

A portable stadiometer (SECA, Hamburg, Germany) was used to measure height while a portable bioelectrical impedance analyzer (TANITA BC 418-MA Tanita Corp., Tokyo, Japan) was used to assess the body weight (kg) and to estimate the whole-body total fat mass, the percentage of body fat, and the fat-free mass. Body mass index was calculated (BMI = weight/height²; kg/m²).

### 2.7. Dietary Intake

A valid semiquantitative food frequency questionnaire [38,39] was used to assess the dietary intake. Daily intake was calculated by multiplying the portion size by the frequency of consumption (9 options ranging from never/almost never to 6 or more times per day). Spanish food composition tables and other sources of information were used to estimate nutrient intake [40,41]. Information collected was relative to the last year. The variables considered in this study were calcium and alcohol due to their relationship with bone and because they were also included in similar studies [20].

### 2.8. Blood Samples and Serum 25(OH)D

Fasting blood samples collected by venipuncture (4 mL). The “VITROS 25-OH Vitamin D Total” package was used to obtain 25OH-D concentrations. This package includes VITROS 25-OH Vitamin D Total Reagent Pack, VITROS 25-OH Vitamin D Total Calibrators, and the VITROS ECi/ECiQ immunodiagnostic system, VITROS 3600. 

### 2.9. Statistical Analyses 

Analyses followed guidelines about compositional data analysis for PA, SB, and sleep research published by Chastin et al. [18]. The R statistical system, version 4.0.2 (Foundation for Statistical Computing, Vienna, Austria; https://www.r-project.org/, accessed on 3 April 2021) was used to perform all the analyses. Men and women were treated as a whole sample in order to have bigger groups and increase the statistical power. As sex showed interaction with all bone variables, it was introduced in all models as a covariate.

First, standard descriptive statistics were performed, indicating results according to the nature of the variables: Continuous (mean ± standard deviation) and categorical (frequency). For compositional data, descriptive characteristics were calculated, including compositional geometric means for central tendency and variation matrices for dispersion. Four ternary plots, each rotating different 3 behaviors, were generated to show the distribution of the sample composition. The overlapped heat map allowed us to distinguish the areas of highest (more intense color) and lowest (less intense color) data concentration and interval confidences. The dispersion structure is represented by normal-based probability regions around the compositional center. This methodology has previously been described by other authors [18,42].

To determine the relationship between movement behaviors and health-related variables of bone mass and structure measured by pQCT, the CoDA was performed. For these analyses, isometric log-ratio data transformation was conducted to adequately adjust the models for time spent in the other behaviors. The combined effects of the relative distribution of all movement behaviors with each outcome were determined by γ and *p*-values, with statistical significance set at *p* < 0.05. The positive or negative associations between each movement behavior and each outcome depending on the time spent in the other movement behaviors were also determined by γ and *p*-values. For those significant variables, further information was described in the text, including the R^2^ value for the whole-movement distribution, and the Standard Error (SE) and *t*-value for individual movements behavior relative to the others. The regression models were adjusted for covariates (sex, age, tibia length, muscle area, smoking, alcohol, calcium intake, and serum vitamin D status) by backward elimination (predictor retained set in *p* < 0.2).

Finally, compositional geometric mean bar plots about absolute proportions of time illustrated the relative movement behavior profiles for fall and fracture indicators (fear of falling, risk of falling, presence of falls in the last year, and presence of fractures during last 10 years).

## 3. Results

### 3.1. Descriptive Characteristics of the Sample

Table 1 shows descriptive characteristic of the sample. From the 103 initial participants, 70 were finally included in this manuscript: 19 men (27%) and 51 women (73%). From the sample included in the present study, 17 individuals were institutionalized. This sample loss was due to the exclusion of participants with poor pQCT image quality or insufficiency accelerometry data. 

### 3.2. Composition of the Day and Movement Behavior Characteristics

The geometric means of the minutes/day and the % of time spent in ST, SB, LPA, and MVPA for the sample are presented in Table 2. 

On average, this sample of older people spent 141.0 min/day in non-sedentary activities. The variability of the data is summarized in the variation matrix containing all pair-wise log-ratio variances, which is presented in Table 3.

As it can be observed in Figure 1, the highest codependences were ST with SB, followed by LPA with MVPA. On the other hand, the lowest codependences were between SB and MVPA, followed by ST and MVPA. The distribution of the sample composition is illustrated in Figure 1 by means of a matrix of ternary plots with three behaviors represented at the same time. Ternary plots can be understood as the scatterplots of compositions [43]. The plot reflects the fact that the highest variability was found in the direction of SB.

### 3.3. Bone Mass and Structure and Movement Behaviors Analysed by Compositional Data

The CoDA models, which show the combined effect of the movement behaviors on each bone variable, are reported in Table 4. The relative distribution of time among the four behaviors as a whole was statistically significantly associated with Ct.Th (R^2^ = 0.253) (*p* < 0.05). MVPA, relative to the time spent in the other movement behaviors, was positively associated with Ct.Th [γST = −0.533 (SE, 0.374 and *t*-value = 0.163), γSB = 0.446 (SE = 0.377 and *t*-value = 1.182), γLPA = −0.502 (SE = 0.375 and *t*-value = −1.336) and γMVPA = 0.590 (SE = 0.201 and t-value = 2.092) (for γMVPA *p* < 0.05)] and Tt.BMC at 38% [γST = −0.288 (SE = 0.187 and *t*-value = −1.535), γSB = 0.383 (SE = 0.191 and *t*-value = 2.002), γLPA = −0.313 (SE = 0.203 and *t*-value = −1.538), and MVPA = 0.218 (SE = 0.103 and *t*-value = −1.538) (for γMVPA *p* < 0.05)].

### 3.4. Composition of the Day by Groups of Fall and Fracture-Related Variables

The composition of the day for each group is presented as compositional mean bar plots. The sample, grouped by having or not having a fear of falling, history of falls, and risk of falls or fractures, is presented in Figure 2 (Figure 2a, Figure 2b, Figure 2c, and Figure 2d, respectively). Older adults who had no fear of falling showed more relative amount of LPA, MVPA, and SB and spent less time in ST compared to the whole sample, while those who reported a fear of falling presented the opposite behavior (Figure 2a).

## 4. Discussion

Some interesting novel findings emerge from this study: (1) Cortical thickness of the tibia length is associated with whole compositional movement distribution in which MVPA, relative to the other movement distribution, showed positive influence; (2) Tt.BMC at 38% of the tibial is positively associated with MVPA relative to the other movement distribution behaviors; (3) fear of falling and previous fractures seem to be associated with reduce levels of LPA and MVPA; and (4) risk of fall and previous falls are related with more PA movement distribution compared with the whole sample.

To the best of our knowledge, this study is the first to use compositional analysis to examine the association between relative distribution of time spent in ST, SB, LPA, and MVPA with bone mass and structure in older adults, measured by pQCT, a more in-depth analysis of bone health status and risk of fracture. This statistical method has been pointed out to be better than a traditional approach in order to understand movement behaviors distribution throughout the day [42]. Moreover, we also included the use of pQCT, which is able to evaluate not only bone mass but also volumetric BMD and bone structural parameters, which may contribute to a deeper understanding of the relationship between movement distributions throughout a day and bone health.

Previous to our research, Rodríguez-Gómez et al. developed a similar study using CoDA and measuring bone by DXA in a sample of Spanish older adults [20], where they described some positive associations between MVPA and pelvic, femur, and trochanter BMD in the whole sample [20]. In this line, prospective results presented by these same authors also reveal a similar positive effect in bone with increases in MVPA [21,22]. Similarly, other authors using traditional statistics and analyzing bone with DXA [44] and pQCT [45,46] have found comparable results regarding MVPA. For example, Füzéki et al. reported positive associations between MVPA and periosteal circumferences of the tibia in a large sample of older women. However, they did not find any association between PA and bone-related variables measured by DXA [46]. Johansson et al. also found positive associations between time spent in MVPA and both cortical area and trabecular BMD in a large sample of 70-year-old men and women [45]. According to all the latter results, older adults may benefit from MVPA even when they spend only little time in this activity intensity. This positive association observed between cortical bone and MVPA implies that the mechanical properties of the bone would be beneficial [47]. Preventing cortical bone loss would lead to delayed bone fragility and protect against fractures and trabecular damage. Thus, MVPA seems to be of high importance in this population in order to improve bone health and prevent osteoporosis- related problems. Nonetheless, the inconclusive results about the influence of LPA and bone variables were presented in a previous systematic review carried out in adults and older adults [48] and, similarly to the unclear associations in our study and the results of our colleagues [20], suggested that LPA could not be enough to improve bone in older adults. Though PA is considered an essential stimulus for bone osteogenesis and its protective action might reduce the effects of bone resorption, it seems necessary to recommend MVPA as it seems to produce higher stimulus than LPA [49]. Hence, our results are not in concordance with the advice that “just being on your feet” is good for health [50] regarding the case of bone. In this case, intensity is a key point to obtain the desired benefits. However, we have to be cautious because even if it does not have a direct effect on the bone parameters, LPA seems the best strategy to limit or displace sedentary time.

In fact, older adults spent too much time sitting, laying, or sleeping, as shown in our results. SB not only negatively affected bone health in other studies [43], but was also related to cardiovascular diseases [51] and frailty [52] independently of the amount of PA, and SB also represents an important risk factor for mortality [53]. However, in our study, SB did not seem to be associated with bone health. This could be partially explained because of the huge amount of time spent in this behavior as other authors has explained [20]. Another possible reason could be that we considered the ratio between SB among the other movement behaviors rather than purely SB time, and we also found a low codependence between MVPA and SB. In this line, the results may change if we consider nonosteogenic activities such as SB and ST together, as the latter showed a negative relationship with bone health in the regression analysis.

Nevertheless, due to the individual effects of PA and SB on overall health, the new PA guidelines recently published by the World Health Organization include, for the first time, the reduction of SB with the aim of improving health in this population [50]. Thus, our study leads to recommend on the line of limiting sedentary time and increasing PA, regardless of the intensity, although MVPA is preferred in relation to bone health. In addition, possible future research lines may focus on the effects of sedentary breaks [54] on bone health, as it is a new approach which may attenuate rest time deleterious effects [55]. Additionally, to analyze not only ST as in the present study and as other authors have done [56,57], but also the quality of this time, seems to be necessary.

Increases of PA levels and high impact exercise have been promoted to reduce the risk of falls, fear of falling, and number of falls and fractures [58]. In relation to the fear of falling, its etiology is still unknown. Although fear of falling usually happens in older adults with previous falls and bone fractures, previous studies have also reported a high prevalence of fear in non-fallers [59,60]. Fear of falling has been previously associated with reduced PA levels [13], as it was the case with our sample. A reduction in PA could negatively impact bone as we have observed and, therefore, predispose individuals to a higher risk of fractures if a fall happens. Moreover, our results also showed reduced levels of PA in those with previous fractures, which may lead to an endless loop, which needs to be stopped by implementing strategies to increase PA. In contrast, those who have suffered a fall in the last year and those at risk of falling showed higher levels of PA, results that are in line with previous studies [12]. Taking into account that the risk of falling depends on having previously suffered a fall, a possible explanation of these results may be that the probability of suffering a fall increases with more time in movement, as a fall does not only depend on the intrinsic capacity of the people but also on external elements such as modifiable environmental factors [61]. In order to avoid possible fractures derived from PA, efforts should focus on safe practice, in which expert-supervised exercise programs seem to be one of the best options in this population group. Concretely, the implementation of multitask and strength exercise has been demonstrated to be effective in the reduction of falls and fractures [62,63,64].

Therefore, our results regarding the distribution of time in movement behaviors showed that those who had suffered a fracture and those with fear of falling need to be the target population in order to increase PA and exercise policies. This beneficial effect of PA in bone health would lead to the prevention of osteoporosis and related fractures both by demising the risk of fall and the possible fractures if falls happen, which represents a major public health problem among older people because it leads to mortality and loss of independence. Hence, taking into account our results, exercise interventions and PA individual recommendations should be designed differently for both of these groups than for those with previous falls or those at risk of falling, as they do not have the same movement behavior patterns. Exercise programs including balance, strength, and specific exercises simulating daily activities would have an extra benefit in all these groups, specially reducing fear of falling [65]. If the fear of falling is reduced, it is likely that PA would be increased and, consequently, some benefits on bone mass may be achieved, as we have shown in our results. Meanwhile, if those fallers and those at risk of falling are encouraged to perform this type of activities, they would decrease probability of suffering a fall and having a subsequent fall, which seems to be likely as they showed patterns with more PA.

Some strengths and limitations of this study should be highlighted. The present study has a cross-sectional design, reflecting associations but not revealing causality. Further research, including larger sample sizes, is required to verify these results in representative populations, and also to allow separate consideration of age and sex groups, which may be influenced differently. Although accelerometry is an objective and validated method, it may not detect differences between sitting and standing positions and may overestimate sedentary time. However, some strengths such as harmonized assessments, well-instructed researchers, objectively measured movement behaviors, volumetric bone measurements, and inclusion of high-quality bone images should also be considered, as well as the novel and comprehensive statistical approach and the inclusion of objectively measured ST. Finally, many of the variables that could have an impact on bone mass were included in the analysis as covariates to ensure that our results were not influenced by these factors (sex, age, tibia length, muscle, calcium, vitamin D, alcohol, and smoking).

## 5. Conclusions

In conclusion, our study showed that cortical thickness is associated to whole movement distribution behavior. Specifically, total true BMD at 38% of the tibia and the cortical thickness are positively influenced by MVPA. Moreover, similar movement distributions were found in those with previous fractures and those with fear of falling. Both groups showed less PA (both LPA and MVPA). In contrast, those who were at risk of falling or with previous falls during the last year showed more active distributions. These results allow us to recommend MVPA for direct positive effect on bone to improve bone structure and to prevent osteoporosis or future fractures. We also recommend general and light PA to replace sedentary time, which is negatively associated with bone tissue and many other health risks.

## Figures and Tables

**Figure 1 ijerph-18-03757-f001:**
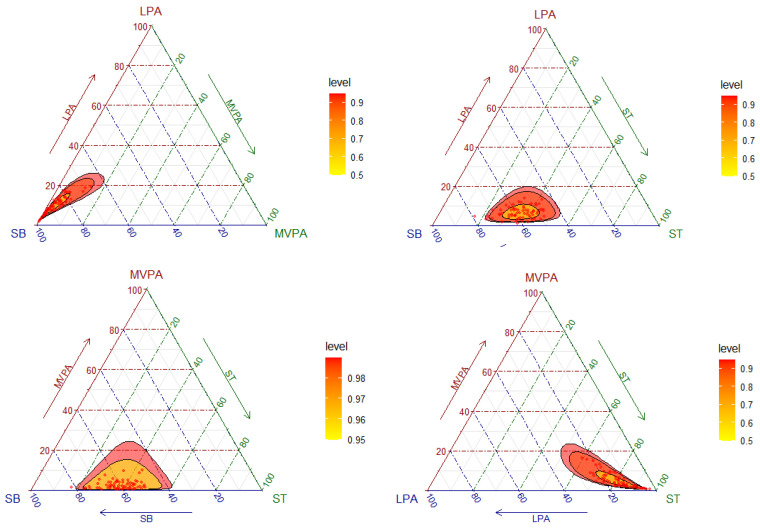
Ternary plots of the sample compositions of time spent in sleep time (ST), sedentary behavior (SB), light physical activity (LPA), and moderate-to-vigorous physical activity (MVPA).

**Figure 2 ijerph-18-03757-f002:**
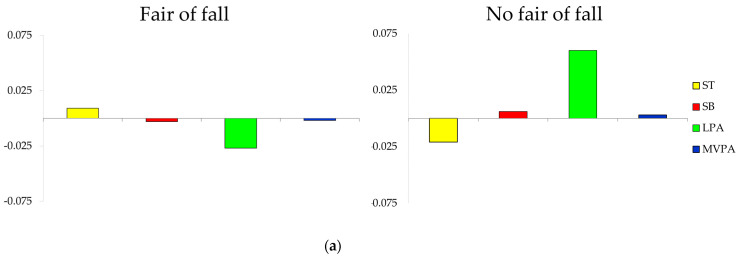
Compositional analysis of the relative importance of the group mean time spent in ST (sleep time), SB (sedentary Behaviors), LPA (light physical activity), and MVPA (moderate-to-vigorous physical activity) with respect to the overall mean time composition of (**a**) those who had and did not have a fear of falling, (**b**) those who had and had not suffered falls in the last year, (**c**) those who were and were not at risk of falling, and (**d**) those who had or had not suffered fractures in last 10 years.

**Table 1 ijerph-18-03757-t001:** Descriptive variables.

Variables	Whole Sample (*N* = 70)
Age (years)	80.4 ± 6.4
*Body composition*	
BMI (kg/m^2^)	28.9 ± 4.8
Body Fat %	37.8 ± 6.7
Tibial Muscle Area	5491.9 ± 1090.6
*Bone health-related confounders*	
Smoking	3(3.8)
Alcohol (g)	4.3 ± 7.5
Serum Vitamin D (ng/dL)	28.5 ± 16.9
Calcium (mg/day)	1123.7 ± 398.5
*Fall-related variables*	
Fear of falling	51(64.6)
Risk of fall	28(35.4)
Falls	32(40.5)
Fractures	54(68.4)
*Bone variables*	
Tt.BMC 4%	2.6 ± 0.7
Tt.BMD 4%	229.4 ± 48.0
Tt.Area 4%	1405.9 ± 199.6
Tb.BMD 4%	203.8 ± 41.6
Tt.BMC 38%	4.1 ± 0.3
Tt.BMD 38%	781.3 ± 109.1
Tt.Area 38%	387.8 ± 59.7
Ct.Th 38%	4.3 ± 0.8
Crt.BMD 38%	122.3 ± 50.5
SSIp	1406.2 ± 339.4
Fracture Load	5133.4 ± 1432.9

Number of participants of the sample (*N*) and % per group for categorical variables, mean and standard deviation (SD) for continuous variables. BMI: Body Mass Index; Fear of falling: People with fear of falling; Risk of fall: People who have risk of having a future fall; Falls: People who had suffered any fall in the last year; Fractures: People who had suffered any fracture in the last 10 years; Tt.BMC: Total bone mineral content; Tt.BMD: Total bone mineral density; Tb.BMD: Trabecular bone mineral density; Tt.Ar: Total bone mineral area; Ct.BMD: Cortical bone mineral density; Ct.Th: Cortical bone thickness; SSIp: Polar stress strain index.

**Table 2 ijerph-18-03757-t002:** Geometric means for SB, LPA, and MVPA in minutes/day and percentage of total wearing hours.

Behaviors	Minutes/Day	% Wearing Hours
ST	507.4	35.2%
SB	827.6	57.5%
LPA	82.3	5.7%
MVPA	22.7	1.5%

ST: Sleep time, SB: Sedentary behavior; LPA: Light physical activity; MVPA: Moderate-to-vigorous physical activity.

**Table 3 ijerph-18-03757-t003:** Pair-wise log-ratio matrix for ST, SB, LPA, and MVPA.

Behaviors	ST	SB	LPA	MVPA
ST	0.000	0.098	0.331	0.930
SB	0.098	0.000	0.355	0.977
LPA	0.331	0.355	0.000	0.286
MVPA	0.930	0.977	0.286	0.000

SB: Sedentary behavior; LPA: Light physical activity; MVPA: Moderate-to-vigorous physical activity.

**Table 4 ijerph-18-03757-t004:** Compositional behavior model for bone mass variables for the proportion of time per day spent in ST, SB, LPA, and MVPA.

Bone Variables.	Model *p*-Value	γST	*p*-Value	γSB	*p*-Value	γLPA	*p*-Value	γMVPA	*p*-Value
Tt.BMC 4%	0.099	−0.417	0.111	0.399	0.134	−0.257	0.332	0.272	0.069
Tt.BMD 4%	0.689	−8.698	0.727	9.813	0.553	−16.114	0.387	14.998	0.300
Tt.Area 4%	0.099	−0.414	0.110	0.399	0.134	0.257	0.332	0.272	0.069
Tb.BMD 4%	0.689	−8.698	0.727	9.813	0.703	−16.114	0.553	14.998	0.300
Tt.BMC 38%	0.105	−0.288	0.133	0.383	0.052	−0.313	0.132	0.218	**0.041**
Tt.BMD 38%	0.883	0.012	0.965	−0.146	0.585	0.195	0.488	−0.061	0.675
Tt.Area 38%	0.264	21.487	0.297	−12.774	0.530	6.254	0.770	−14.967	0.180
Crt.BMD 38%	0.773	9.937	0.785	8.932	0.793	−36.899	0.328	18.029	0.332
Ct.Th	**0.012**	−0.533	0.163	0.446	0.245	−0.502	0.190	0.590	**0.005**
SSIp	0.477	−150.65	0.120	121.51	0.202	24.466	0.801	4.677	0.926
Fracture Load X	0.700	−371.13	0.420	176.36	0.702	70.65	0.886	124.12	0.627

All models are adjusted for sex, age, object length, muscle area, alcohol intake, smoking, serum vitamin D, and calcium by backward elimination (with predictor retained if *p* < 0.2). Statistically significant associations (*p* < 0.05) are highlighted in bold. ST: Sleep time, SB: Sedentary behavior; LPA: Light physical activity; MVPA: Moderate-to-vigorous physical activity; Tt.BMC: Total bone mineral content; Tt.BMD: Total bone mineral density; Tb.BMD: Trabecular bone mineral density; Tt.Ar: Total bone mineral area; Ct.BMD: Cortical bone mineral density; Ct.Th: Cortical bone thickness; SSIp: Polar stress strain index.

## Data Availability

The data are not publicly available due to privacy.

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
