# Peer review of "Associations between Daily Movement Distribution, Bone Structure, Falls, and Fractures in Older Adults: A Compositional Data Analysis Study"

_ijerph, 2021, doi:10.3390/ijerph18073757_

Round 1
Reviewer 1 Report
The QCT exam involves a higher dose of ionizing radiation than a DXA exam. The authors should convince me of the need for QCT compared to DXA in the context of their research. After all, the main conclusion of the work presented by the authors is that the total BMD at 38% of the tibia is positively influenced by MVPA. The authors must explain why it may also be important to point out that cortical thickness at 38% of the tibia is positively influenced by MVPA.
More details about the data regression would be useful to better understand the value of the results.
I suggest improving the quality of figure 1 because it is unreadable
Reviewer 2 Report
Dear sir, the work appears to be a part of a very broad research project by the authors. The study was approved by "ethics committee from Hospital Universitario Fundacion de alcorcon", but the study was carried out in other city or region. None of the authors appear as member of that hospital or of any associated research center. This is an importan irregularity. Especially in studies where the study population is subjected to significant radiological exposure, rigorous control is necessary. This aspect should be clarified.
Data on study populatios, its selection etc must be clarified. The authors refer to a previous study (ref 26) that is not easily accesible to understand the methodology. The characteristics (biodemographics, clinics, treatments etc) of the population play a important role. The choice of the tibial bone should be explained. Other authors have used the same bone, but it is true that other bone segments are the most affected by osteoporosis and changes in bone mineral density. The differences between men and women have already been highlighted in previous studies. The problem that a significant physical activity (MVPA) can entail in the most fragile age groups should be discussed.
Reviewer 3 Report
The article is well structured. The introduction explores the results of other studies and allows you to understand why this study was carried out.
The method and techniques are clear and presented with rigour.
The results are pertinent and the discussion allows for a deeper understanding of them.
I think it can be published in the current format.
Round 2
Reviewer 2 Report
Thanks, the work has improved.